# Exploring Sketch-based Character Design Guided by Automatic Colorization

Rawan Alghofaili*
George Mason University

Matthew Fisher
Adobe Research

Richard Zhang
Adobe Research

Michal Lukáč
Adobe Research

Lap-Fai Yu
George Mason University

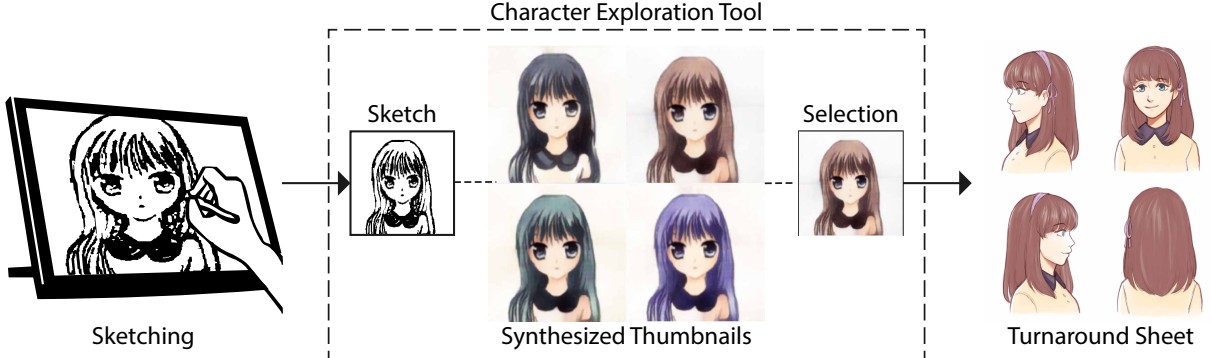

Figure 1: Our character exploration tool facilitates the character design process (more specifically, character exploration) by allowing artists to explore characters using colored thumbnails synthesized from sketches. These colored thumbnails, which are traditionally rough grey-scale sketches, better visualize the character for creating the turnaround sheet. The turnaround sheet was created from the selected thumbnail by Ruba Alhumaidi.

## ABSTRACT

Character design is a lengthy process, requiring artists to iteratively alter their characters' features and colorization schemes according to feedback from creative directors or peers. Artists experiment with multiple colorization schemes before deciding on the right color palette. This process may necessitate several tedious manual re-colorizations of the character. Any substantial changes to the character's appearance may also require manual re-colorization. Such complications motivate a computational approach for visualizing characters and drafting solutions.

We propose a character exploration tool that automatically colors a sketch based on a selected style. The tool employs a Generative Adversarial Network trained to automatically color sketches. The tool also allows a selection of faces to be used as a template for the character's design. We validated our tool by comparing it with using Photoshop for character exploration in our pilot study. Finally, we conducted a study to evaluate our tool's efficacy within the design pipeline.

**Index Terms:** Human-centered computing—Human computer interaction (HCI)—Interaction paradigms—Graphical user interfaces

## 1 INTRODUCTION

Fig. 1 illustrates a typical character design process. At the very beginning of the process, the designer is furnished with a character description that outlines a combination of personality (e.g., courageous, melancholic) and physical traits (e.g., long hair, small frame) [7, 30]. Their first task is then to sketch out the character's distinguishing expressions and physical features into a thumbnail—which is often a rough low-resolution gray-scale sketch. From the thumbnail the designer then develops a character turnaround sheet, a reference for later drawing the character in context. The turnaround

---

*e-mail: ralghofa@gmu.edu

sheet is then presented to the creative director for feedback, and the entire process iterates. Because the ideation and creation of a turnaround sheet are manual processes, the artist often has to restart from scratch.

We devised a tool driven by a sketching interaction that automatically colors the character thumbnail, enabling artists and their creative directors to do more early exploration with less investment of effort. In practice, these thumbnails may also be used as references for producing the turnaround sheet in higher resolution using Photoshop. Fig. 1 shows an example of a colored character thumbnail synthesized using our tool.

Our novel tool can generate these colored character thumbnails based on artists' sketches, color and character face selections. Specifically, we achieve this by training a Generative Adversarial Network (GAN) using an anime dataset. We used the GAN to generate the colored thumbnails as the artists sketch, while also allowing them to place characters' faces and select their colorization style. This selection-based automatic colorization framework was able to significantly speed-up the character exploration process compared to using Photoshop for participants in our user study, without sacrificing quality. The major contributions of our work include:

- Proposing a novel generative character exploration tool by training a GAN to automatically color sketches.

- Allowing artists using our tool to select character faces and colorization style, as well as to edit the character by directly sketching on the canvas.

- Validating the effectiveness of our tool in facilitating the character exploration process compared to Photoshop via a number of design tasks.

## 2 RELATED WORK

### 2.1 Sketch-based Interactions

Similar to our approach, several works have explored using sketching as an interaction technique in different contexts.

We draw inspiration from several works that utilized sketching as an animation interaction. Kazi et al. [24, 25] created an interface to allow users to animate their 2D sketches, while Guay et al. [11] presented a novel technique to animate 3D characters' motion using a single stroke. Storeoboard [15] allows filmmakers to sketch stereoscopic storyboards to visualize the depth of their scenes.

Our approach aims to incorporate sketching into a 2D design process, while several works aim to examine sketch interactions in 3D design. Saul et al. [36] created a design system for chair fabrication. Xu et al. [48] introduced a model-guided 3D sketching tool which allows designers to redesign existing 3D models. Huang et al. [16] created a sketch-based user interface design system. ILoveSketch [3], a curve sketching system, allows designers to iterate directly on their 3D designs. Sketch-based interaction techniques in Augmented Reality were explored by the HCI community as well [2, 28, 43].

Several works explored using sketching to design cartoon characters specifically. Sketch2Manga [32] creates characters from sketches. Unlike our approach which uses a generative method to output a character from a sketch, it uses image retrieval to match the query with a character from the database. Han et al. [12] introduced a deep learning method to create 3D caricatures from an input 2D sketch. Because the generated caricatures take the form of a texture-less 3D model, we opted to use a network architecture that enabled the generation of 2D images and control of their colorization style.

With our tool, we aim to improve the traditional design process for artists. Similarly, Jacobs et al. [19] introduced a tool which allows artists to create dynamic procedural brushes by varying the rotation, reflection and style of their strokes. Moreover, Vignette [26] is an interactive tool which allows artists to create custom textures, and automatically fill selected regions of their illustrations with these textures.

## 2.2 Image Generation

Recently, generative modeling approaches have emerged as a powerful, data-driven approach for directly mapping sketches into images. Isola et al. [18] show that conditional GANs are an effective general purpose tool for image-to-image translation problems and can be applied to mapping sketches to images. The sketch-to-image problem is also inherently ambiguous, as different colors and "styles" can be used for multiple plausible completions. Follow-up works [17, 49] introduce extensions to enable multiple predictions. We find that for our task BicycleGAN [49] is able to effectively generate colored character illustrations from edge maps due to its multimodality. One challenge is the difficulty in obtaining real sketches. Some methods [5, 9, 39] use generative models to generate sketches themselves. We find that "synthesized" sketches based on edge maps, with some carefully selected preprocessing choices, are adequate for our application.

Using the style selector in our tool, artists can choose the colorization scheme of their characters. Similarly, Color Sails [38] is a tool which allows coloring designs from a discrete-continuous color palette defined by the user. Tan et al. [41] developed a tool to allow real-time image palette editing. Zou et al. [50] introduced a language-based scene colorization tool. Xiang et al. [46] explored the style space of anime characters by training a style encoder which effectively encodes images into the style space, such that the distance of their codes in the space corresponds to the similarity of their artists' styles.

In later work, Xiang et al. [47] developed a Generative Adversarial Disentanglement Network which can incorporate style and content codes that are independent. This allows separate control over the style and content codes of the image, enabling faithful image generation with proper style-specific facial features (e.g., eyes, mouth, chin, hair, blushes, highlights, contours,) as well as overall

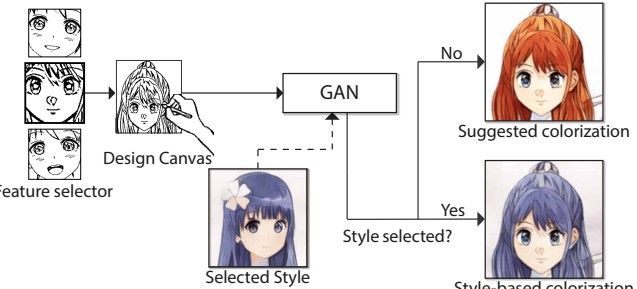

Figure 2: Overview of our approach. To begin, an artist may place a face from the face selector onto the design canvas. The artist then directly sketches on the design canvas. If a style is selected, the sketch will be automatically colored by the GAN with the selected style. Otherwise, the tool suggests a random colorization scheme.

color saturation and contrast. The neural transfer methods used by Xiang et al. [46, 47] do not transfer facial features consistently (e.g., may transfer the mouth from some images but not all). Therefore we allow artists to control facial features via only the sketch canvas and/or face selector instead of using the neural transfer methods of Xiang et al. Nonetheless, due to its effectiveness, we still used neural transfer to control the sketch's colorization.

## 2.3 Character Design

EmoG [37] is a character design tool introduced to facilitate storyboarding. EmoG generates facial expressions according to the user's emotion selection and sketch. Akin to our approach users can drag and drop a facial expression onto the canvas in addition to the ability to draw directly on the canvas. Unlike our approach, EmoG renders no colorization suggestions to the user and is focused on facilitating drafting characters' emotional expressions rather than their overall appearance.

MakeGirlsMoe [20] is a tool that helps artists brainstorm by allowing them to select facial features to automatically generate a character illustration. However, it has an unnatural discrete selection-based interaction compared to interfaces that allow the user to illustrate by sketching. MakeGirlsMoe was updated to create the crypto-currency generator, Crypko [6]. Both frameworks were not available to us during the user evaluation and thus were not compared to our tool. PaintsChainer [34] automatically colors sketches based on the artist's color hints in the form of brush strokes on top of the sketch. It colors a completed line art that a user uploads, though it does not allow the user to modify the character by placing or editing expressions and features onto the canvas, nor does it allow the user to start from a blank canvas and iteratively sketch a character. Consequentially, it neglects the need for a sketch-based iterative tool that combines both a feature selection-based interaction and automatic colorization. Hence, we developed an interactive character design tool equipped with a face selector, colorization style selector and sketching canvas to fulfill that need.

Auto-colorization features were introduced in commercial software like Adobe Illustrator and Clip Studio Paint. However, Adobe Illustrator is limited to coloring black-and-white photographs. On the other hand, Clip Studio Paint can color cartoons, but like PaintsChainer, it can only color completed line art.

## 3 OVERVIEW

Fig. 2 shows an overview of our approach. We trained a GAN by using an edges-to-character dataset obtained by extracting the edgemaps of colored anime characters. The GAN learned to produce a colored anime character illustration given a sketch. Using the GAN we built a framework which allows character exploration by enabling a user to select and place facial features as well as sketch onto a

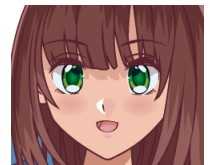 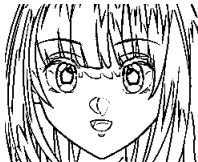 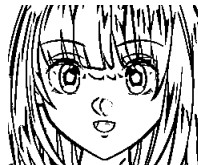

| (a) Character Image | (b) $\sigma = 0.3$ | (c) $\sigma = 0.5$ |

Figure 3: To synthesize artist sketches we used an edge operator on our dataset images. (a) A character image sampled from our training set. The edgemaps were created by applying the DoG filter with (b) $\sigma = 0.3$ and (c) $\sigma = 0.5$.

canvas. As the user edits the canvas, the GAN will automatically color his illustrations according to the styles he selected. Finally, we demonstrated the effectiveness of our tool by conducting a user study comparing our tool with Adobe Photoshop.

## 4 DATA PROCESSING

We obtained our training and validation image pairs from the anime-face character dataset [33]. We used an automated process described below to extract edge maps from the face images, creating our edges-to-character dataset.

**Animeface Dataset.** The animeface character dataset [33] contains a total of 12,213 samples of face images. We randomly extracted 10,992 images of them for the training set. The remaining 10% of samples (1221 images) were used as validation to monitor the progress of training the GAN.

**Edgemaps.** Due to the costliness of obtaining character datasets which include sketches paired with their corresponding colored counterparts, we used an edge detector on the dataset images to simulate sketches. The standard Difference of Gaussians (DoG) filter was used successfully in several works to synthesize line drawings [10,21,29,45], and unlike the eXtended difference-of-Gaussians (xDoG) filter [44] it does not tend to fill dark regions. Before processing the images, we created the edgemaps of our training and validation images after converting them to grayscale and then applying the DoG filter with $\gamma = 10^9$ and $k = 4.5$ (see Fig. 3). The value of $\sigma$ was randomly selected from $\{0.3, 0.4, 0.5\}$ for each image to allow for variations in the amount of noise in the edgemaps.

**Image Processing.** The images in the animeface dataset have a maximum size of 160px in either dimension and various aspect ratios, while our GAN training process expects images sized exactly $256 \times 256$px. In order to match these requirements, we uniformly scaled the animeface faces to fit using bilinear interpolation. For other than square aspect ratios, we filled the rest of the square canvas by repeating edge pixels as shown in Fig. 4. We selected this repetition fill rather than a solid background color to avoid the network learning to reproduce such a solid border.

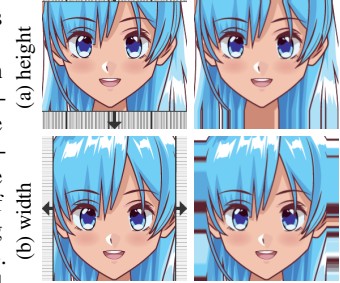

Figure 4: The arrow depicts the direction of replicating the border pixels when the (a) height or (b) width is the smallest dimension.

## 5 CHARACTER COLORIZATION MODEL TRAINING

To generate each colored character image from its paired edgemap image in our dataset, we used BicycleGAN from Zhu et al [49].

**Architecture.** We train the network on the $256 \times 256$ paired images from our training edges-to-character dataset. For our encoder, we found that using a ResNet [14] encoder explored by Zhu et al. helped

decrease the amount of artifact images generated by the GAN. We use a U-Net [35] generator and PatchGAN [18] discriminators.

In preliminary experiments, we found that changing the dimension of the latent code $|\mathbf{z}|$ produces different results. A code of too high dimension leads to variation in the background style instead of the character colorization style, while too low dimension leads to inadequate variation in the character colorization style. Ultimately, we found a latent code of size 8 to empirically work well. GANs are known to "collapse" when training lasts too long [4]. Subsequently, we noticed that eventually colorization resolution improves at the expense of style variation after 71 epochs as the GAN starts over-fitting on the training data. Because our tool is created to explore character designs and produce low-resolution sample thumbnails, we opted to halt training at 71 epochs to maintain variation in style and to avoid overfitting.

**Training.** We inherit many of the default parameters and practices of BicycleGAN: $\lambda_{\text{image}} = 10$, $\lambda_{\text{latent}} = 0.5$ and $\lambda_{\text{KL}} = 0.01$. We trained for 71 epochs using Adam [27] with batch size 1 and learning rate 0.0002. We updated the generator once for each discriminator update, while the encoder and generator are updated simultaneously. We used the TensorFlow library [1]. Training took approximately 48 hours on an Nvidia GeForce GTX 1070 GPU.

We found these parameters empirically. Note that our goal is not to produce the state-of-the-art generative model for this task per se, but rather to explore how a reasonable implementation of a powerful generative model can be leveraged for downstreaming character exploration by an artist.

## 6 CHARACTER EXPLORATION TOOL

Due to its multi-modality, our trained neural network is able to color each edgemap in various styles. We demonstrate the several methods incorporated in our character design tool shown in Fig. 6 to color character sketches. The colorization results we presented in this section were generated using the same apparatus used for training.

**Suggested Colorization.** We can color the edgemaps by randomly sampling the latent code $\mathbf{z}$ from a Gaussian distribution and injecting it into the network using the add_to_input method explored by Zhu et al. [49], which spatially replicates and concatenates $\mathbf{z}$ into only the first layer of the generator.

Fig. 5 shows colorization results of images in our validation set by randomly sampling the latent code. By varying the latent code the network was able to vary the character's hair color. Because the majority of anime characters in the dataset have matching hair and eye colors, the network jointly varies the hair and eye color. Darker hair colors can be generated by increasing the amount of shading as can be seen in the second row of Fig. 5.

**Style-based Colorization.** We can also inject the latent code $\mathbf{z}$ of other images (i.e. style images) into the network, which enables us to color the input edgemap according to the style images. We first encode the style image to its latent code $\mathbf{z}$. We then generate the character image from the edgemap by injecting the style image's latent code $\mathbf{z}$ using the add_to_input method.

Fig. 7 shows the results of coloring input edge images from our validation sets using a set of style images. Due to the inclusion of multi-faced images within the training set, the network is able to color multiple faces in one image (as illustrated by the final row of Fig. 7), giving artists the ability to sketch multiple faces on the same canvas. These faces are generated with the same colorization scheme. Because we did not remove the backgrounds from the training images, the GAN generates the backgrounds as part of the image's style.

**Implementation Details.** We designed the character exploration tool (shown in Fig. 6) to allow artists to sketch on the design canvas using the brush and eraser provided. The brush is circular and its diameter can be adjusted using the brush slider from 1 to 10 pixels.

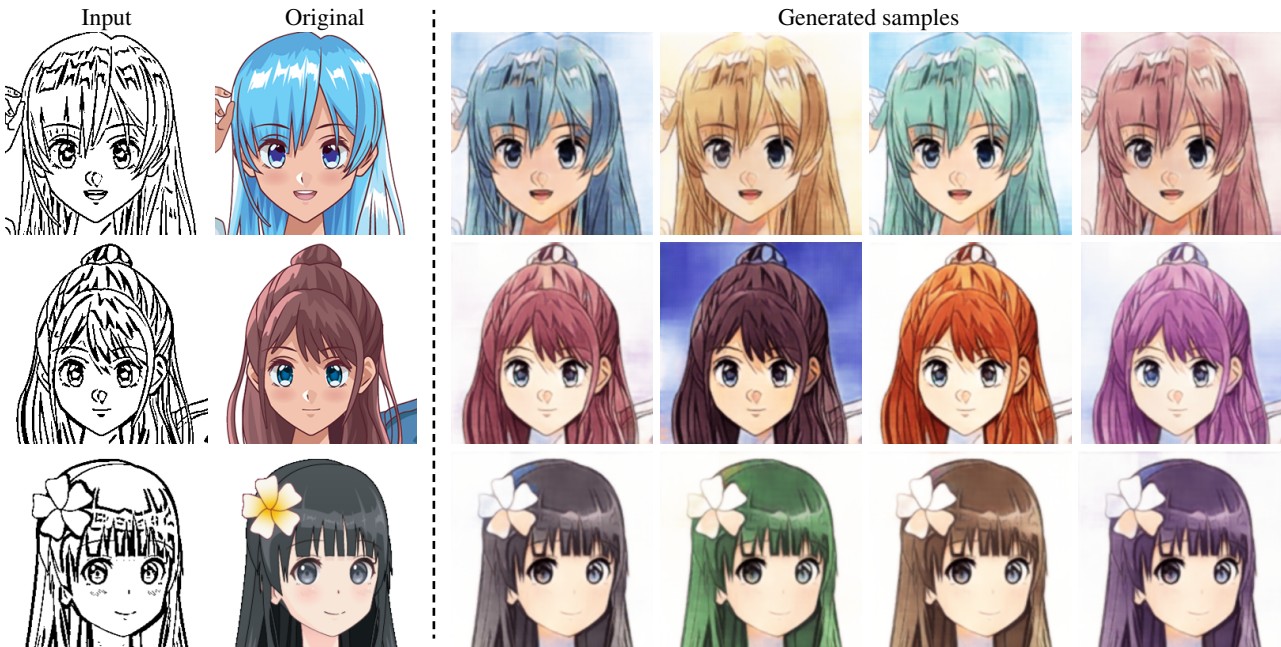

Figure 5: Sample suggested colorizations from our model. The first column shows the input edgemap. The second column shows the original image. The last four columns show the colorization results of our network with a latent code **z** randomly sampled from a Gaussian distribution for each generated sample.

The eraser is likewise circular and its diameter can be adjusted in the range of 1 to 20 pixels.

Artists are also able to place facial expressions into any location on the design canvas from our face selector by clicking the facial expression and the canvas respectively. The face selector provides artists with templates of the most common anime faces, which helps artists visualize the direction they want to take the design prior to fully drawing it out. The facial expressions were created by extracting the faces detected by applying an anime face detector [40]. We selected 60 of the faces detected in the validation set to be used in the face selector.

The style selector provides a set of style images from our validation set. These style images were selected by embedding the 8 dimensional latent codes of images in our validation set into two dimensions using t-SNE [31]. The embeddings are visualized as a 10x10 grid by snapping the two dimensional embeddings to the grid. The embeddings were arranged in the grid such that every position in the grid contains the style image with the latent code which has the smallest Euclidean distance to the grid position. Twelve images of the 100 style images visualized using t-SNE embedding were discarded due to the presence of some artifacts after using them as style images in colorization. Therefore, we used 88 images in total in our style selector. For consistency and to avoid the varying background, resolution and artistry of the style images from biasing artist's selections in the user study, we display preview images colored with the style images shown in the t-SNE grid using the style-based colorization method. Fig. 6 shows some of these preview images in the style selector. Please refer to the supplementary material for the t-SNE grid visualization.

The colored sketch will be shown on the display canvas. If the artist has not selected a style image in the style selector, the image will be colored using the *suggested colorization* method. Otherwise, the sketch will be colored according to the *style-based colorization* method using the artist's selection in the style selector as the style image. The display canvas will be automatically updated every 20 seconds. The update can also be triggered by the artist by pressing the *run* button. If a style image was not selected, pressing the *run* button will trigger applying random colorization with a newly

sampled latent vector, giving the artist an additional way–other than the style selector–to explore the colorization space. The sketching canvas can be cleared by pressing the *clear* button.

## 7 PILOT USER STUDY

**Participants.** We recruited 27 artists with ages ranging from 19 to 30 to participate in our IRB-approved study. Fig. 8 shows participants' average years of experience with sketching (M=5.52, SD=4.65), character design (M=2.26, SD=2.96) and using Adobe Photoshop (M=3.26, SD=3.04). Participants' experience is listed in more detail in the supplemental material.

**Setup.** Participants sketched on a Wacom Cintiq Pro 13 tablet with a 13-inch display. Our tool was loaded on the tablet. The participants sketched directly on the screen. We used the same apparatus employed in training the GAN to generate the images of the display canvas.

**Tasks.** Following the completion of a training task, participants were given 6 tasks. Each design task refers to a combination of design request, time condition, and tool condition. Participants completed each of the 6 design requests shown in Table 1, which were created under the consultation of a professional character designer. The time conditions *Limited* and *Unlimited* determined whether participants completed the request within a 15-minute limit or under no time constraints, respectively. The *Limited* time condition was used to compare the quality of designs under tight time constraints. Our tool conditions are defined as: our character design tool, our character design tool supplemented with pencil/paper, and Adobe Photoshop condition. To allow for within-subject comparisons between tool conditions under each time condition, participants completed all of the 3 tool conditions under each time condition. The ordering and combinations of the design requests, tool conditions and time conditions were randomized for each participant to avoid any carryover effects. For example, one participant may be given the first design request from Table 1 to be completed under the Adobe Photoshop and *Unlimited* conditions as their first task; while another participant completes the third design request under the character design tool and *Limited* conditions first. On average, participants completed the study–including the training task–in approximately

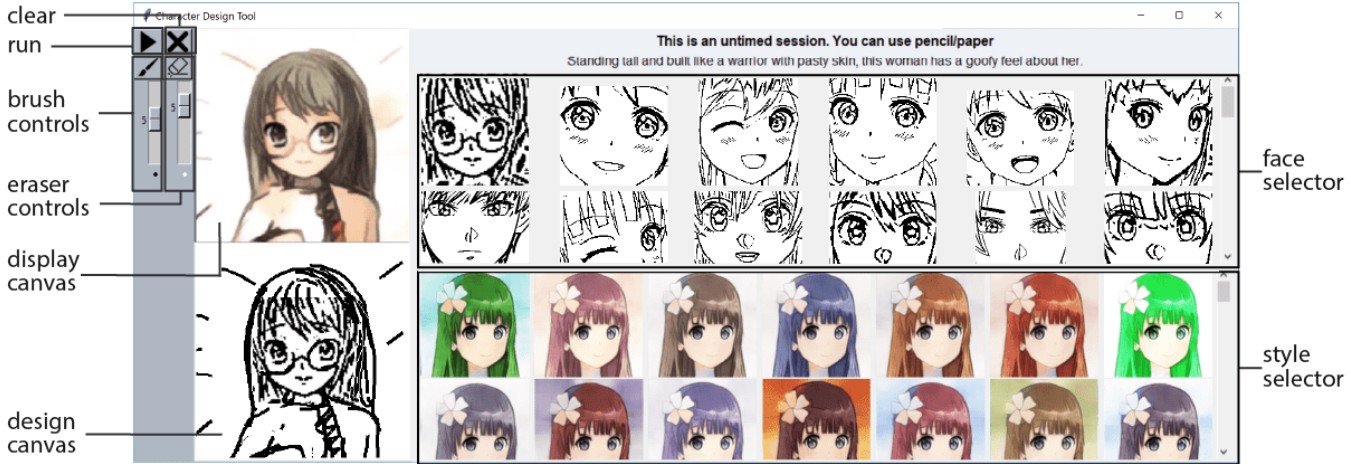

Figure 6: The UI of the character exploration tool in our user study. The canvases show a thumbnail designed using our tool.

| Input | Original | Style 1 | Style 2 | Style 3 | Style 4 | Style 5 | Style 6 |

Figure 7: Style-based colorization using our model. The left column shows the input edgemap while the second column shows the original image. The six rightmost columns show the results of style-based colorization on the edgemap with various styles.

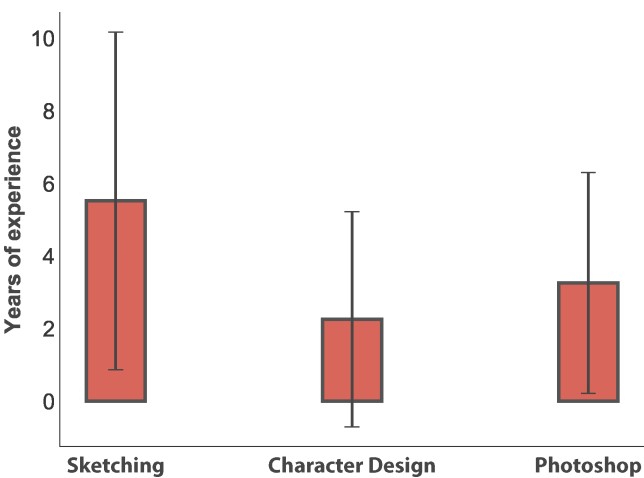

Figure 8: The participants' average years of experience with sketching, character design, and using Adobe Photoshop.

| | Design Request |
|---|---|
| D1 | Cheerful female character with long hair. She has a cool and flowing appearance. |
| D2 | She's a cold, lone wolf with a sense of humor. |
| D3 | A determined and patient girl with a simple and practical look. Her greatest desire is ultimate knowledge. |
| D4 | She's a determined and courageous healer, with a dark and eerie appearance. |
| D5 | She's a dedicated and knowledgeable scholar with a bright and sunny aesthetic. |
| D6 | She's a charming and fun-loving socialite with a vintage and classic look. |

Table 1: The 6 character design requests given to participants in our user study.

90 minutes.

### Training Task

We allowed participants to freely explore the tool before receiving any design requests. To facilitate the learning process, we provided participants with a tutorial explaining the various functionalities of our tool along with the study's structure.

### Character Design Tool

Participants completed two design requests using our tool and without using a pencil or paper. One request was completed within 15 minutes while another was completed without a time constraint.

### Character Design Tool and Pencil/Paper

Participants were allowed to use a pencil and paper for two design requests completed using our tool. They completed one request within 15 minutes, and one without a time constraint. Some artists typically plan their designs on paper prior to using editing software. Thus, we added this tool condition to inspect whether allowing artists to use pencil/paper affects their workflow.

### Adobe Photoshop

Similar to the previous tool conditions, participants completed two requests by using Adobe Photoshop, once without a time constraint and once with a 15-minute limit. To mimic participant's typical design process as closely as possible, we provided them with a pencil and paper in this tool condition as well.

**User Survey.** After completing the 2 tasks under each tool condition (i.e. once with a 15-minute time limit and once without),

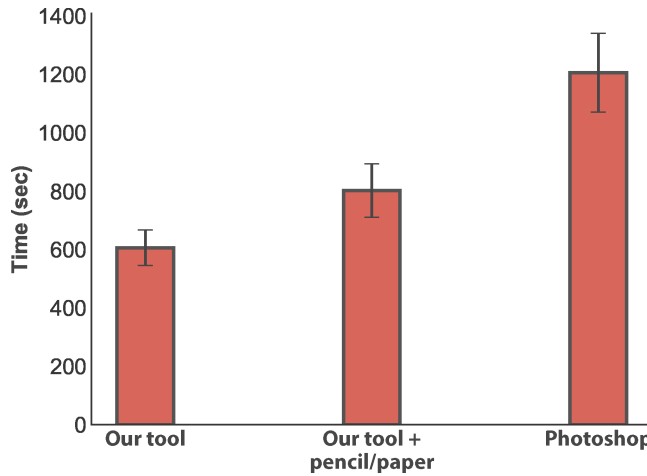

Figure 9: Average time of completing the design requests.

participants were asked to complete a survey to evaluate the performance of the tool used. We opted to use a 5-point Likert scale to evaluate the tools akin to [24]. Participants were asked to evaluate the following statements with a rating of 1 (strongly disagree) to 5 (strongly agree):

- The tool was easy to use and learn.

- I find the tool overall to be useful.

Finally, we surveyed participants once more after completing the user study in its entirety. Participants were asked to rate each of their colored designs from 1 (Poor) to 5 (Excellent). Participants are aware of the study's time constraints, so they are more likely to fairly judge their artworks' quality. Therefore, we opted to rely on the participants' evaluation of their own work instead of using external evaluators. We were also interested in learning which designs participants favored overall, so for each time condition the participants were asked to vote for either the design created under the *Character Design Tool*, *Character Design Tool and Pencil/Paper*, or *Photoshop* condition.

### 7.1 Time of Completion.

Fig. 9 shows the average time taken by participants to complete the character design requests under each tool condition. Mauchly's test did not show a violation of sphericity against tool condition ($W(2) = 0.84$, $p = 0.11$). With one-way repeated-measure ANOVA, we found a significant effect of the tool used on the time of design completion ($F(2,52)$=14.53, partial $\eta^2$=0.36, $p < 0.001$). We performed Boneferroni-corrected paired t-tests for our post-hoc pairwise comparisons.

Participants completed the designs faster by using our tool compared to Photoshop. A post-hoc test showed that the average time participants took to complete the designs using Photoshop ($1205 \pm 135.16$ seconds) was longer than using our tool with pencil/paper ($801.7 \pm 91.28$ seconds) ($p = 0.01$). A post-hoc test also showed that participants completed the design requests in a shorter amount of time by using our tool without pencil/paper ($605.93 \pm 60.73$) ($p < 0.01$) compared to Photoshop. The post-hoc test showed no significant difference in the time of completion when comparing completing the task using our character design tool with or without pencil/paper ($p = 0.12$). This suggests that while using our tool sped-up the design process when compared to Photoshop, including the pencil/paper did not yield any observable significant improvements in our setting.

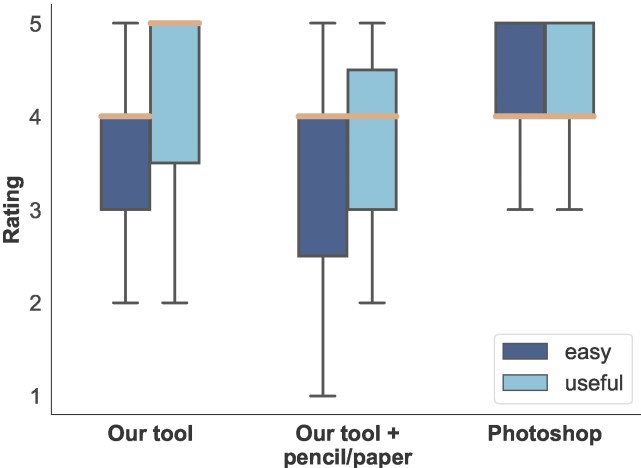

Figure 10: Participants answered the questions in the experience survey with a rating of 1 (strongly disagree) to 5 (strongly agree)

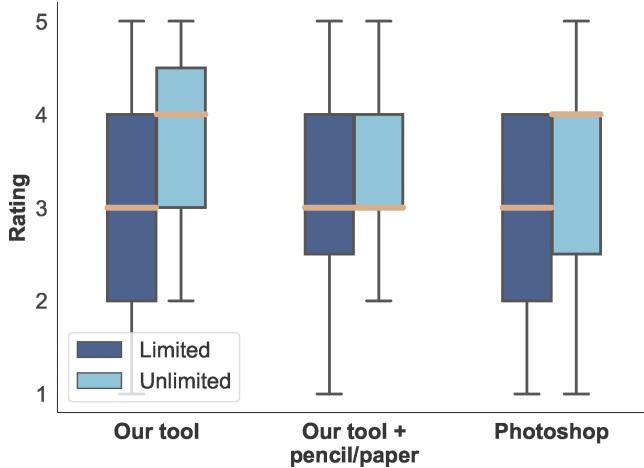

Figure 11: Participants were asked to rate their designs with a rating of 1 (poor) to 5 (excellent). *Limited* and *Unlimited* refer to whether the design was created with a 15-minute time limit or with unlimited time.

Participants remarked that our tool expedites the design process (*P1,P3,P10,P20*). *P3* specifically noted that our tool "*makes producing a character design much faster and easier than doing it on paper.*"

## 7.2 Evaluation of Experience Survey

Fig. 10 shows participants' response to "*The tool was easy to use and learn.*" for each tool condition. A Friedman test showed a significant difference in participant's responses to the statement ($\chi^2(2) = 13.73$, $p = 0.01$). We also conducted post-hoc analysis using Wilcoxon signed-rank tests with Bonferroni correction. Similar to Adobe Photoshop, the median of participants found our tool easy to use (Md=4 agree). However, the post-hoc tests showed a significant difference between the ease of use of our tool and Adobe Photoshop. In other words, we found a significant difference when comparing participants' responses after using our tool without pencil/paper and Adobe Photoshop ($W = 199, Z = -3.04, r = 0.41, p = 0.007$). Likewise, we found a significant difference when comparing using our tool with pencil/paper and Adobe Photoshop ($W = 144, Z = -3.25, r = 0.44, p = 0.003$). These results may be observed in Fig. 10 by the broader variation in responses given to our tool conditions compared to the *Photoshop* condition.

The familiarity of photo editing software to our participants may have contributed to the consensus of Adobe Photoshop's ease of use compared to our tool. Although some participants like *P26* appreciated the simplicity of our application by stating that "*it's modestly easy to use for character designers of any experience level. It's perfect as it is.*", the absence of exhaustive common features that exist in modern editing software might have contributed to our tool's wider range of easiness ratings. The post-hoc test showed no significant difference between the ease of using our tool with or without pencil/paper ($W = 33, Z = 0.92, p = 0.35$).

Our Friedman test found a significant difference in participants' responses to the "*I find the tool overall to be useful.*" statement as well ($\chi^2(2) = 9.86$, $p = 0.007$). The post-hoc test ($W = 63.5, Z = -1.33, p = 0.56$) showed no significant difference between the usefulness rating of using Adobe Photoshop (Md=4 agree) compared to using our tool without pencil/paper (Md=5 strongly agree), despite our tool having a higher median rating than Adobe Photoshop. Conversely, the post-hoc test ($W = 135, Z = -2.86, r = 0.39, p = 0.013$) showed a significant difference between the rating of Adobe Photoshop and using our tool with pencil/paper despite having the same median rating (Md=4 agree). Furthermore, we found no significant difference between responses under the *Character Design Tool* (Md=5 strongly agree) and *Character Design Tool and Pencil/Paper*

(Md=4 agree) conditions ($W = 4, Z = -2.49, r = 0.34, p = 0.038$). The inclusion of pencil and paper as an additional step in the participants' pipeline might have made the design process more cumbersome, resulting in the tendency to view the usefulness of our tool under the *Character Design Tool and Pencil/Paper* condition to be less than the other two conditions as shown in Fig. 10.

## 7.3 Evaluation of Designs

Fig. 11 shows how participants rated the designs produced using the various tool conditions we studied. The designs produced under the *Limited* constraint were rated similarly (Md=3) under all the tool conditions. A Friedman test also indicated no significant difference in the rating of designs produced under that time constraint ($\chi^2(2) = 1.98, p = 0.37$).

Although the median of ratings was higher for images designed under the *Character Design Tool* and *Photoshop* conditions (Md=4), than the *Character Design tool and Pencil/Paper* condition (Md=3); we found no significant difference in the rating of designs produced without any time constraints by applying the Friedman test ($\chi^2(2) = 4.13, p = 0.13$).

Some participants (*P4,P12*) noted that the artwork they produced during the user study does not reflect their abilities. This may suggest that the participants may be rating the designs based on their previous body of work, giving all the designs overall a neutral rating; consequently resulting in no significant difference in the rating of images under different tool conditions. Nevertheless, the designs created using our tool received the majority of participants' votes as can be seen in Fig. 14.

Fig. 12 shows some selected participant's thumbnails using our tool, while Fig. 13 shows their designs using Photoshop. The examples created under the *Character Design Tool* condition seem to be of better quality than their *Photoshop* counterparts. The participant also created the design faster by using our tool (386 seconds) compared to using Photoshop (620 seconds) while under the *Unlimited* time condition. Although the designs created using Photoshop are comparable to ones created under the *Character Design Tool and Pencil/Paper* condition, the time it took to complete the design using our tool (388 seconds) was much shorter for the participant than using Photoshop (652 seconds) while under the *Unlimited* time condition. The remaining thumbnails are included with the supplemental material.

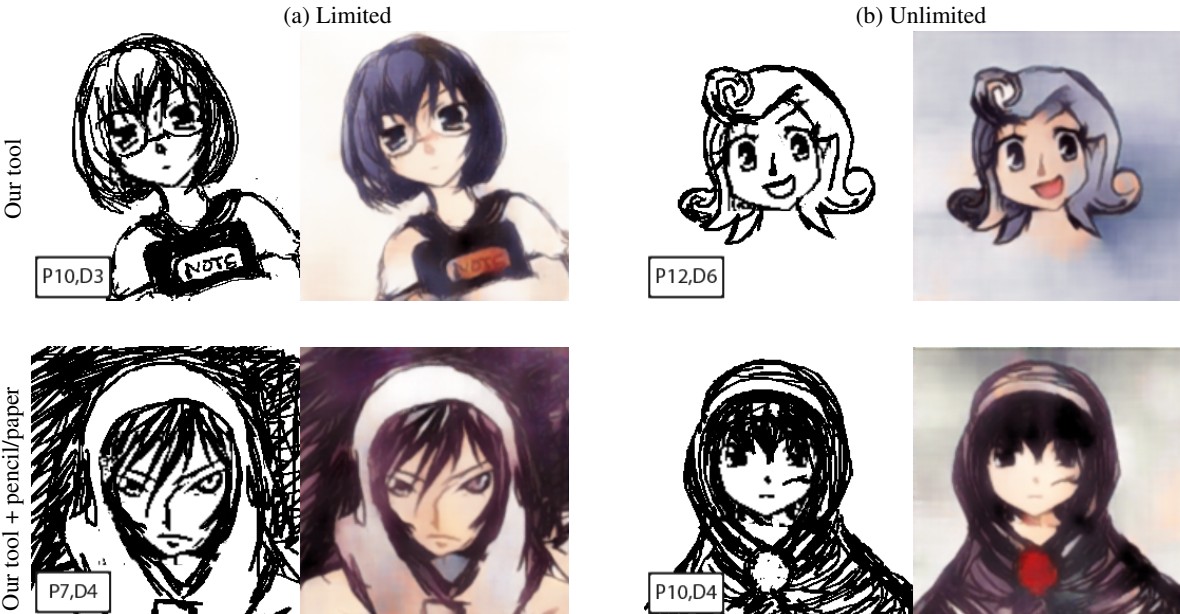

Figure 12: Participant's sketches and their corresponding colored thumbnails created using our tool under the (a) *Limited* time condition and the (b) *Unlimited* time condition. The images were labeled with the participant number and design request completed (D3:*"A determined and patient girl with a simple and practical look. Her greatest desire is ultimate knowledge."*; D4:*"She's a determined and courageous healer, with a dark and eerie appearance."*; D6:*"She's a charming and fun-loving socialite with a vintage and classic look."*). *P10* used the face selector to design the character according to D3 and D4, while *P7* used the face selector to design the character according to D4.

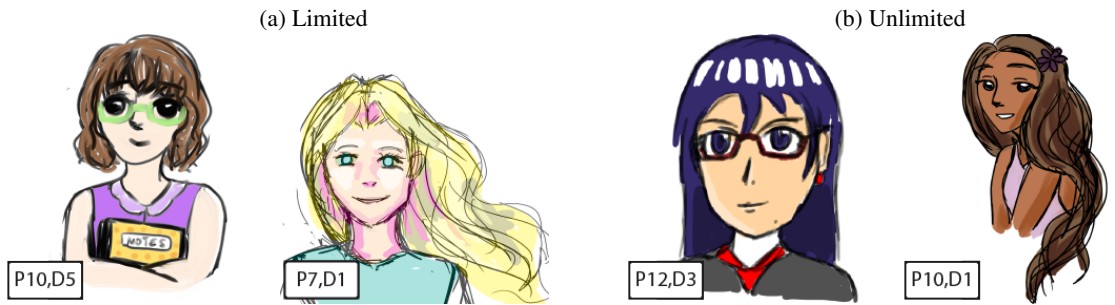

Figure 13: Characters created by the participants from Fig. 12 using Photoshop. The two leftmost illustrations were created under the (a) *Limited* time condition, while the two rightmost were created under the (b) *Unlimited* time condition. (D1:*"Cheerful female character with long hair. She has a cool and flowing appearance."* ; D3:*"A determined and patient girl with a simple and practical look. Her greatest desire is ultimate knowledge."*; D5:*"She's a dedicated and knowledgeable scholar with a bright and sunny aesthetic."*).

## 8 EVALUATION OF THE TOOL'S USAGE IN THE WILD

To evaluate the effectiveness of our tool in the design workflow, we conducted a user study that simulates directors' and artists' workflow in the character design process. Due to the pandemic, we were unable to recruit a large number of participants and thus conduct a large-scale user study. Moreover, our user study was conducted remotely.

**Participants.** We recruited 5 of the artists with ages ranging from 19 to 30 in our initial user study to participate in our second IRB-approved study. We also recruited 5 participants of ages 19-25 to act as art directors.

**Setup.** To use our tool the artists were asked to connect to the same device utilized in our initial user study using TeamViewer. For comparison, the artists were asked to use their preferred drawing

tools. We placed no constraints on the software the artists used. Instead, we encouraged artists to employ the tool that will most facilitate the brainstorming process for them. Some artists used tools that have auto-colorization capabilities like Adobe Illustrator and Clip Studio Paint, while others selected tools that did not support auto-colorization like FireAlpaca and PaintTool SAI. The artists, directors, and researcher used Zoom to communicate.

**Tasks.** Before the study, each director submitted two character designs. Two different artists were randomly assigned to each director to complete his/her designs. The directors introduced their designs to each artist in a brainstorming session. Moreover, the artists shared their screens in these sessions to show their sketches to the directors. The artists used our tool in one brainstorming session and their selected drawing tool in the other. The session was terminated when

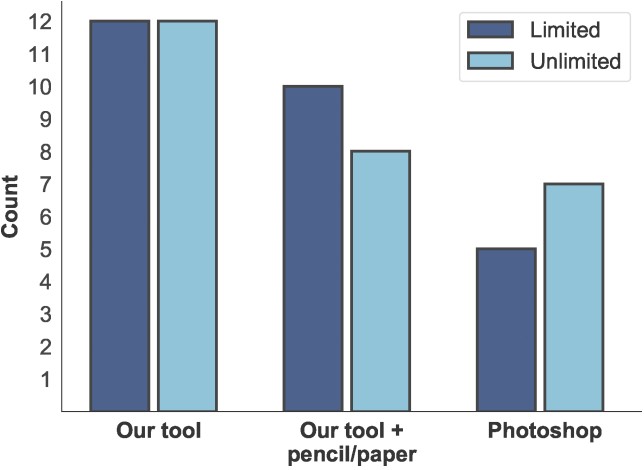

Figure 14: The number of participants (out of 27) who selected the designs of highest quality created under each tool condition.

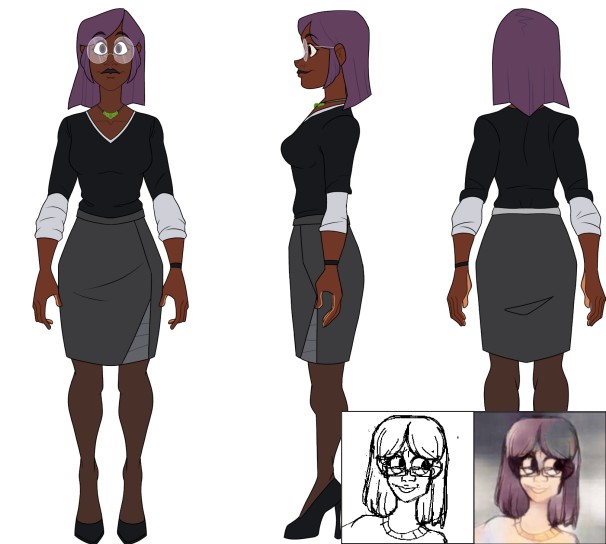

Figure 15: The turnaround sheet created by *Artist 1* after using our tool in the brainstorming session. The lower right corner shows the colored thumbnail produced by our tool and its input sketch.

the director was satisfied with the rough design the artist produced. The time of completing these sessions was recorded to compare our tool with other drawing tools.

After the brainstorming session ended, the artists submitted the turnaround sheets to their respective directors via e-mail. The artists iterated on the designs based on the directors' feedback. The study concluded when the directors approved the turnaround sheet that each of the two assigned artists submitted.

After the directors approved an artist's turnaround sheet, they were asked to complete a survey. The directors were asked to evaluate the following statements with a rating of 1 (strongly disagree) to 5 (strongly agree):

- I am satisfied with the quality of design the artist produced.

- The design matches my description.

- Communication with the artist in the brainstorming session was easy.

The artists were asked to report the number of hours they spent working on the design, as well as to evaluate the following statements with a rating of 1 (strongly disagree) to 5 (strongly agree):

- Communication with the director in the brainstorming session was easy.

**Results.** The amount of time in the brainstorming session is shown in Table 2 as *Session time*. On average, the artists spent a shorter amount of time in the brainstorming session by using our tool ($855 \pm 95.02$ seconds) compared to using other drawing tools ($1584.8 \pm 165.29$ seconds). Despite artists spending a shorter amount of time in the brainstorming sessions, they spent a similar amount of time working on the designs after using our tool ($3.1 \pm 0.9$ hours) in the brainstorming session compared to after using other drawing tools ($3.2 \pm 0.86$ hours).

Moreover, directors overall were satisfied with the quality of the turnaround sheets produced by the artists after using our tool (Md=5 strongly agree) akin to after using other drawing tools (Md=5 strongly agree). Directors overall felt that the turnaround sheets produced after using our tool (Md=4 agree) matched their description as well. The turnaround sheets created in our user study are included in the supplementary material.

Only one director was unsatisfied with the turnaround sheet the artist produced after using our tool in the brainstorming session. The

director worked with *Artist 1*, giving a score of 2 (disagree) to both the quality of the design and its match to the director's description. In the brainstorming session, the artist produced the thumbnail shown in Fig. 15 which the director approved. In the e-mail correspondence after the brainstorming session, the director was indecisive about the character's specifications. These miscommunications resulted in both the artist and director rating the communication in the brainstorming session lower than in any other session, with the artist giving the director a score of 1 (strongly disagree) and the director giving the artist a score of 3 (neutral) as can be seen in Table 2.

Overall, both artists and directors believed that communication with their counterparts went smoothly. The directors rated communication with the artists using our tool (Md=5 strongly agree) akin to using other drawing tools (Md=5 strongly agree) during the brainstorming session. Artists reported slightly better communication with the directors after using our tool (Md=5 strongly agree) compared to other drawing tools (Md=4 agree).

## 9 DISCUSSION

**Limitations.** Although participants believed that our tool allows them to draft a character much faster than Photoshop, they encountered some limitations to the framework. For example, although our tool was able to color multiple faces sketched by an artist within the same canvas, our GAN tends to style all faces with the same color scheme, limiting designs to only one character per canvas. Moreover, *P3* suggested to *"simply use the facial expressions, as opposed to the expressions plus some of the hair"* in the face selector. Due to the face detection method we utilized, the selections we provided in the face selector included some portions of the characters hair. With a more sophisticated feature segmentation and classification model, the different facial features (e.g., hair, eyes) could be segmented and displayed separately in the selector. Some participants expressed the need for improvements to the interface like a larger sketch canvas (*P7*), an undo button (*P3, P4, P7, P11, P13, P14, P19, P27*), and stroke sensitivity/customization (*P3, P5, P9, P12, P14*).

Our style selector is not fully customizable. For example, *P3* wanted the ability *"to have a way to set up a custom hair and skin color."* Using an architecture similar to the one proposed by Karras et al. [23] to transfer the style could allow a more finely-grained customization of the colorization scheme.

|  | Artist 1 | | Artist 2 | | Artist 3 | | Artist 4 | | Artist 5 | | Average | |
|---|---|---|---|---|---|---|---|---|---|---|---|---|
|  | Ours | Other | Ours | Other | Ours | Other | Ours | Other | Ours | Other | Ours | Other |
| Session time (seconds) | 1,200 | 1,800 | 900 | 1,860 | 666 | 1,113 | 797 | 1,893 | 712 | 1,258 | 855 | 1584.8 |
| Creation time (hours) | 3 | 3 | 1 | 1 | 1.5 | 2 | 6 | 6 | 4 | 4 | 3.1 | 3.2 |
| Quality | 2 | 5 | 5 | 5 | 5 | 5 | 5 | 5 | 5 | 5 | 5 | 5 |
| Description matching | 2 | 5 | 5 | 5 | 4 | 5 | 5 | 3 | 4 | 5 | 4 | 5 |
| Communication (director) | 3 | 5 | 5 | 5 | 5 | 5 | 5 | 5 | 5 | 5 | 5 | 5 |
| Communication (artist) | 1 | 5 | 5 | 2 | 5 | 4 | 5 | 3 | 5 | 5 | 5 | 4 |

Table 2: Results of comparing our tool to other drawing tools in our second user study. Our tool's results are shown in the *Ours* columns while other tools' are shown in the *Other* columns. The *Session time* indicates the duration of the brainstorming session in seconds. The *Creation time* indicates the number of hours the artists spent working on the design after the brainstorming session. The *Quality* indicates how the directors rated their satisfaction with the quality of the turnaround sheet. *Description matching* indicates the director's rating of how well the turnaround sheet matched their description. *Communication (director)* indicates the rating of communication during the brainstorming session that the director reported, while *Communication (artist)* indicates the rating the artist reported.

While some participants were content with the variety of faces (*P9, P11, P26*) and styles (*P14, P18, P27*) our tool provides, due to limitations in our dataset, our tool does not provide artists with large variations in skin tone, nor does our tool provide a substantial number of non-female characters in the face selector for the same reason. Our tool is able to color male characters as illustrated in Figure 7 which we also provide in the face selector. Moreover, participants like *P7* who, despite the usage of female pronouns in the design description, created non-female characters using our tool (as shown in Fig. 12). However, they identified the need for further inclusivity, especially in the face selector (*P7, P21*). Nevertheless, 20 out of the 27 participants used the face selector in at least one of their final designs.

Participants overall praised the face selector in expediting the design process by providing a baseline for the character (*P1, P2, P3, P6, P10, P26, P27*). *P27* found the face selector *"made the app more useful in comparison to photoshop because you could start out with a template"*.

We received positive feedback from user study participants regarding our tool's applicability within the design pipeline (*P1, P3, P6, P9, P10*). Moreover, incorporating aspects of the NASA TLX (Task Load Index) [13] could also aid with further investigating our tool's usability.

**Future Work.** The current focus of our tool was to expedite the exploration of character faces. By expanding our dataset to include full-body character images we may be able to train a network with an architecture similar to Esser et al.'s [8] to generate characters in various poses, body types and clothing. This may expand the capabilities of our tool to aid artists in creating the entire turnaround sheet in addition to exploring the character's head-shot. PaintsChainer [34] allows artists to provide color hints for the tool. We may be able to achieve a similar interaction by including hint channels into our GAN's input layer. The GAN can be trained by randomly sampling colored strokes from the character images in the edges-to-character dataset and using them as the inputs to the hint channels.

Expanding our dataset to more styles of characters may also allow us to cater to designers who have a style dissimilar to anime as requested by *P*10 and *P*21. Fig. 16 shows a participant's design using Photoshop compared to our tool. Although our GAN could detect and generate portions of the character's hair and skin, the result is less than optimal when compared to the participant's design with Photoshop. The GAN in this case fell short of differentiating some portions of the skin (e.g., character's neck) from shading or determining the borders of the character's hair precisely.

Participants believed that our tool was effective in creating images which can be used in the character exploration phase of the design process (i.e. thumbnails) but not as finished pieces (*P1, P12*). Expanding its capabilities to generate high-resolution images (as

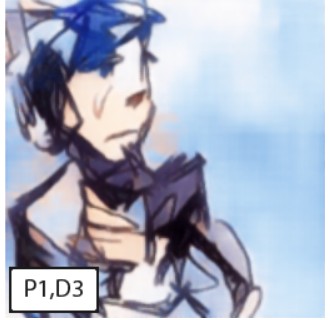 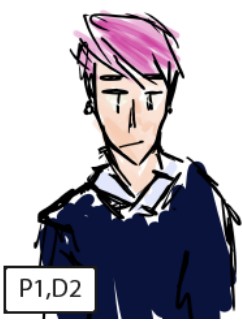

| (a) Our tool, Limited | (b) Photoshop, Unlimited |

Figure 16: A character design which strayed from our dataset's anime style. (a) The participant created the design under the *Character Design Tool* and *Limited* time conditions without using the face selector. (b) The participant created the design under the *Photoshop* and *Unlimited* time conditions. (D3:*"A determined and patient girl with a simple and practical look. Her greatest desire is ultimate knowledge."*; D2:*"She's a cold, lone wolf with a sense of humor"*)

suggested by *P12*), textures, lighting, and shading may broaden its applicability from a simple brainstorming tool to a standalone design tool. We may be able to achieve a higher-resolution output by modifying our GAN's architecture and training it with high-resolution images, or by using the method proposed by Karras et al. [22] to train our GAN with low-resolution images. Finally, removing the generated background may produce more polished finished pieces. We may be able to remove the generated backgrounds in post-processing by training a semantic segmentation network [42] to label backgrounds which can be subtracted thereafter. Alternatively, we may be able to suppress the GAN from generating backgrounds by subtracting the backgrounds from our dataset, and training the GAN with the background-removed images.

## 10 CONCLUSION

In this paper, we trained a Generative Adversarial Network (GAN) to automatically color anime character sketches. Using the GAN we created a tool that aids artists in the early stages of the character design process. We evaluated the efficacy of our tool in comparison to using Photoshop by conducting a user study, which showed our tool's potential in speeding-up the character exploration process while maintaining quality. Finally, we conducted a user study that simulates the director and artist interaction in the design pipeline. We concluded that our tool facilitated character design brainstorming without sacrificing the quality of the designs.

## ACKNOWLEDGMENTS

We are grateful for the anonymous reviewers for their constructive feed back. We thank Atheer AlKubeyyer and Mazen Almusaed for helping us formulate our problem statement. We would especially like to thank Ruba Alhumaidi for creating the artwork for this paper and regularly providing feedback to improve our prototype. The DCXR lab acknowledges the generous support of Adobe through unrestricted gifts. This project was supported by an NSF CAREER Award (award number: 1942531).

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
