# OpenReview forum: "Exploring Sketch-based Character Design Guided by Automatic Colorization"
_graphicsinterface.org/Graphics_Interface/2021/Conference/Second_Cycle — GI 2021_

### Official Review · Reviewer_eQCw · 2021-04-19
**The authors present a sketch interface, based on suggestions of a GAN for colorization. The main contribution is a user study evaluating this interface. The paper is well written and easy to follow. Hence, I suggest acceptance.**

**Rating:** 8
**Confidence:** 2

**Review:**

This is a paper proposing a sketch interface that allows the user to (a) choose a template of a face and (b) pick from different colourization options driven by a GAN. The authors perform two user studies, one testing out the speed and quality of the interface and a second one understanding how the interface supports collaboration.

As far as I can tell, the technical realization makes sense and the user studies have been well performed. Further, the paper is easy to read and follow. Hence, I would suggest acceptance of the paper.

The only issue of concern that I found was that the questions asked in the user survey (page 6) to have the danger of confirmation bias. It would be better to use the SUS or NASA=TLX here.


However, there are a number of minor aspects, that should be fixed before publishing.

* "Methods such as [5, 9, 40]" references shouldn't be used in that fashion. A sentence should change it's meaning if references are left out.
* "euclidean" should be capitalized (it's a name)
* "Fig. 8 shows participants’ average experience with sketching (M=5.52, SD=4.65), character design (M=2.26, SD=2.96)" ... at this point there is no unit and therefore no meaning attached to these numbers
* Fig 9 should be reporting in terms of minutes (ie. human readable number)
* Fig 10: "(strongly agree" misses a ')'
* I don't understand Fig 14. What is meant with "selected the design"? what was the constraint / setup to the question to the participants?

The authors should do a major revision of their references. Almost every ref needs some fixing:
- there are too many arxiv/corr papers listed: please reference the accepted papers
- many papers have a wrong capitalization of their title: 2, 3, 4, 5, 8, 12, 13, 23, 29, 32, 33, 35, 38, 42, 44, 45,
- many references have a wrong capitalization of the journal title: 3, 13, 17, 18, 25, 32, 47,
- some papers miss important publication detials: 14, 21, 28, 30, 48, 49, 50, 51
- ref 17+18 are identical

---

### Official Review · Reviewer_hA6f · 2021-04-29
**The paper presents a character exploration tool for character designing, which employs automatic coloration based on a generative adversarial network (GAN).**

**Rating:** 4
**Confidence:** 3

**Review:**

The tool is designed to alleviate the workload of the tedious manual (re-)coloration that may happen in the iterations in the character designing process. The tool focuses on the exploration of the character thumbnail, which consists of a character face exhibiting the traits of the character used to make detailed character design later. The tool interactively updates the coloring based on the sketch drawn by the artist and the selection of the coloring styles. Also, the tool is capable of choosing the face template and starting the design process based on the template. To validate the effectiveness of the approach, the authors conducted user studies comparing the approach against the workflow without the proposed tool.

The paper presents a character exploration tool for character designing, which employs automatic coloration based on a generative adversarial network (GAN). The tool is designed to alleviate the workload of the tedious manual (re-)coloration that may happen in the iterations in the character designing process. The tool focuses on the exploration of the character thumbnail, which consists of a character face exhibiting the traits of the character used to make detailed character design later. The tool interactively updates the coloring based on the sketch drawn by the artist and the selection of the coloring styles. Also, the tool is capable of choosing the face template and starting the design process based on the template. To validate the effectiveness of the approach, the authors conducted user studies comparing the approach against the workflow without the proposed tool.The paper reads smoothly and I was able to clearly understand the claim by the author.  However, I have several concerns that explain my rating as follows.

## Technical novelty
Unfortunately, I am not convinced by the technical novelty of the system asserted in the paper. The network architecture for the auto-colorization is inherited from BicycleGAN [Zhu et al. 2017] without justification for the selection of the architecture. The recommended styles are merely a visualization of latent space using the t-SNE grid, which is also a common approach. Overall, I felt that the system is a collection of the known techniques, which gave me an impression that the technical contribution is marginal.

## Scope
The paper should include a discussion about the scope of the approach. The system focuses on helping the character designing process, but surely it does not support the entire process since the tool mainly focuses on creating the character’s face. For instance, the system is not capable of designing physical traits related to the whole body, nor a detailed design of clothing or ornaments, which is often important to characterize the figure.  At the first sight, I had an impression that the coloring of the turnaround sheet is also automated, but it is not.

## Variation of styles
To me, the variation of the styles (e.g., shown in Fig. 5, 7) seems merely a change of hair colors within a single art style. I wonder what contributes to this outcome. I expected the tool to support a more vibrant change of styles, like various combinations of colors for the base color, shade, highlight, etc.  If the change of color only matters, applying color filters to the layer corresponding to the air color would be sufficient.

## Face template
I didn't get the purpose of the face template since I could not imagine the situation that a professional artist uses this feature to design a character. Using a facial template means an artist uses the other's art style (in the data set) and the fixed facial expression. I wonder if it is a feasible assumption for the artist since a face tends to be an important figure in the character design reflecting the aesthetic taste of the artist. Replacing the process surely reduces the exploration time to fix the design, but it also means the variation of the style and the facial expressions are limited to the styles of the template, which may diminish the originality of the work.

## Evaluation
From the experiment in Sec. 7, I felt the time measurement and the quality of the image are largely influenced by the face selector. Since the artist can start the drawing based on the template, it enables the artists to completely skip the time to draw the same part of the face.  Also, since the template is selected from the anime data set, it already has a stable art style, which should contribute to the quality of the sketch. I suspect it explains why the drawings using the proposed tool generally have better quality. Along this line, I'm curious about the comparison when the face selector is disabled.

I want to point out the feature of the face selector is nothing but using a template from a fixed set of images as a starting point of the design. This means you can do the same process with Photoshop.

Although the tool can improve the early exploration phase of the character design, I'm not convinced that the approach is useful due to the limited applicability. This impression is also exhibited in the result from Sec. 8. Indeed, the tool could improve the efficiency of brainstorming, the time was amortized by the creation time which takes much longer than the time for the brainstorming.

## Conclusion
Overall, I'm not convinced by the usefulness of the approach. My rating reflects the concerns discussed above.

## Minor comments
- Fig. 1. The art styles between the image created by the tool and the turnaround sheet are too different. Is it intentional? It seems the hair color and the shape of the collar are inherited from the selected image.
- p.3 right. "due to the inclusion of multi-faced images within the training set" Is this statement correct? The data set is supposed to contain a single anime face. It may contain images with multiple faces but in most cases, it is cut off at the border.
- p.4 left. It is better to specify discard 12 images in the supplemental material.
- p.4 left. Checking the supplemental material, the shown t-SNE grid doesn't seem to capture any continuous change of the style in the grid. It seems mostly random. Is there any reason to explain this behavior?
- p.4 left. It is better to indicate the runtime for a single coloring operation.
- Fig. 6. I'm confused by this figure. The images shown in the style selector are not the same as the images in the supplemental material. What are the images shown in the style selector exactly?
- Fig. 6,7. The images used in the figures are not images included in the data set, right?
- Fig. 12. It seems all of the participants used the face selector but it doesn't match the caption.
- Sec. 7. It is better to include all images drawn by the participants in the supplemental materials and the selected images in the paper.
- Sec. 7. Is any restriction of the features in Photoshop? The drawing capability of the proposed tool is limited, I wonder a valid comparison is possible without restricting the feature of the tool.
- Sec. 8. Is there any case that a director rejected the submitted turnaround sheet and asked to update it?

## Additional references
- Zhang et al. Style Transfer for Anime Sketches with Enhanced Residual U-net and Auxiliary Classifier GAN. 2017.
- Zhang et al. Two-Stage Sketch Colorization. 2018.
- Ren et al. Two-Stage Sketch Colorization With Color Parsing. 2019.
- Frans. Outline Colorization through Tandem Adversarial Networks. 2017.
- Liu et al. Auto-painter: Cartoon Image Generation from Sketch by Using Conditional Wasserstein Generative Adversarial Networks. 2017.
- Hensman and Aizawa. cGAN-based Manga Colorization Using a Single Training Image. 2017.

---

### Official Review · Reviewer_WaYn · 2021-05-03
**nice paper, very little technical contribution but good user experience**

**Rating:** 6
**Confidence:** 3

**Review:**


This paper presents an interactive method for sketch colorization based on style selection. This is both an important problem in computer animation as well as a well-researched one - unfortunately, not being my primary area of expertise I cannot comment on the completeness of the related work nor on the choice to only compare against photoshop tools.

While I enjoyed reading the paper and I liked the overall idea, the paper has very little technical contribution: the use of GAN for this type of problem has been done in the past, both the generator and discriminator network are off-the-shelf networks. Moreover, no insight was given as to why the authors selected these networks or why these particular networks are best suited for this application. Furthermore, these networks take as inputs 256x256 images and the images used were 128x128 and instead of adapting the networks, the authors chose to naively upsample the images.

On the other hand, I think this paper fits very well the profile of GI, the user study seems thorough, and based on that it seems that this tool could be useful for artists especially if it were to be publicly released. This is the main reason for leaning slightly towards acceptance.

Another few small comments:
* Authors mention that more details about the expertise of the user will be provided in the supplemental material, but I found none
* Also for this type of tool, a video showcasing the tool would be very important, and I would really like to see it as supplemental material is the paper gets accepted.

---

### Meta-Review · Area_Chair_9HoC · 2021-05-07

**Recommendation:** Accept
**Confidence:** 2

**Metareview:**

Despite the weak technical novelty, the reviewers felt rather positive about the method as a tool for character exploration and colorization as well as the evaluation. Taking into account the work as a whole, its fit to the venue and the intended audience, I recommend acceptance if the authors can address the following issues:

•	Better clarify the scope of the paper – it is not clear which steps of the design process are supported by this tool

•	To that end, please add a video as a supplemental material that includes screens-recording of a character design using this tool so that the reader can better understand the scope and its ease of use

•	Explain in more detail the purpose of the template

•	Several missing citations and typos (check the individual reviews)

•	Address the other comments from individual reviews to the best of your ability

---

### Decision · Program_Chairs · 2021-05-08

Accept